# Development of Layer-by-Layer Magnetic Nanoparticles for Application to Radiotherapy of Pancreatic Cancer

**DOI:** 10.3390/molecules30061382

**Published:** 2025-03-20

**Authors:** Nobuyoshi Fukumitsu, Yoshitaka Matsumoto, Lili Chen, Yu Sugawara, Nanami Fujisawa, Eri Niiyama, Sosuke Ouchi, Emiho Oe, Takashi Saito, Mitsuhiro Ebara

**Affiliations:** 1Department of Radiation Oncology, Kobe Proton Center, Kobe 650-0047, Japan; fukumitsun@yahoo.co.jp; 2Department of Radiation Oncology, Clinical Medicine, Faculty of Medicine, University of Tsukuba, Tsukuba 305-8575, Japan; saitoh@pmrc.tsukuba.ac.jp; 3Proton Medical Research Center, University of Tsukuba Hospital, Tsukuba 305-8576, Japan; yu-suga0315@outlook.com; 4Smart Polymers Group, Research Center for Macromolecules and Biomaterials, National Institute for Materials Science, Tsukuba 305-0044, Japan; chenlili03070526@hotmail.com (L.C.); nfujisawa.tokyo@gmail.com (N.F.); lab.sosuke@gmail.com (S.O.); emiho0219@gmail.com (E.O.); ebara.mitsuhiro@nims.go.jp (M.E.)

**Keywords:** layer-by-layer magnetic nanoparticles, radiosensitization, thermo-chemotherapy

## Abstract

Pancreatic cancer is among the deadliest malignancies, with few treatment options for locally advanced, unresectable cases. Conventional therapies, such as chemoradiotherapy and hyperthermia, show promise but face challenges in improving outcomes. This study introduces a novel drug delivery system using gemcitabine (GEM)-loaded layer-by-layer magnetic nanoparticles (LBL MNPs) combined with alternating magnetic field (AMF) application and X-ray irradiation to enhance therapeutic efficacy. LBL MNPs were synthesized using optimized layering techniques to achieve superior drug loading and controlled release. Human pancreatic cancer cells (PANC-1) were treated with LBL MNPs alone, with AMF-induced hyperthermia, and in combination with X-rays. The results demonstrate that the 7-layer LBL MNPs exhibited optimal cytotoxicity, significantly reducing cell viability at concentrations of 30 µg/mL and higher. Combining 7-layer LBL MNPs with AMF increased cell death in a time- and concentration-dependent manner, achieving up to 98% inhibition of cell proliferation. The addition of X-rays to the regimen demonstrated a strong synergistic effect, resulting in a 13-fold increase in cell death compared to controls. These findings highlight the potential of this integrated approach to improve outcomes in patients with pancreatic cancer.

## 1. Introduction

Pancreatic cancer has a poor prognosis, with the annual death toll steadily rising to approximately 38,000 in 2021, making it the third leading cause of cancer-related deaths in Japan, according to Cancer Statistics in Japan [1]. Surgical resection is currently the only curative treatment; however, less than 20% of cases are resectable at presentation. Additionally, up to 25–30% of newly diagnosed patients with non-distant metastatic lesions are classified as borderline resectable or unresectable based on the extent of vascular involvement. In these unresectable cases, chemotherapy or chemoradiotherapy is administered [2]. Despite advancements in drug development and radiotherapy techniques over recent decades, the prognosis for pancreatic cancer remains poor. Therefore, the development of alternative treatments combining chemotherapy and radiotherapy is urgently needed [3].

Hyperthermia has a sensitizing effect on chemotherapy and radiotherapy, potentially improving the efficacy of these therapies for diseases that are highly resistant to treatment. Recent studies have demonstrated that subablative hyperthermia at approximately 42 °C significantly enhances chemotherapeutic effects by increasing the sensitivity of cancer cells to chemotherapeutic drugs and reversing chemoresistance [4,5,6,7,8,9]. Although biological evidence suggests that hyperthermia is effective for the treatment of cancer, its widespread clinical acceptance remains limited. This is primarily due to the challenge of selectively heating the cancer lesion while minimizing effects on surrounding normal tissues, particularly for deep-seated tumors such as pancreatic cancer.

In this study, hyperthermia and chemotherapy, along with various materials and drug-delivery techniques, were developed [10,11,12,13,14,15,16,17]. The layer-by-layer (LBL) method, innovatively introduced by Decher et al., has been extended due to its operational simplicity [18,19]. To achieve precise drug delivery, functional material units must be designed for each component, and carriers that allow flexible control over the structure are highly beneficial. In this study, magnetic nanoparticles were selected as the heat source, and induction heating through irradiation with an alternating magnetic field was utilized. Furthermore, for use in combination with chemotherapy, a polyelectrolyte was coated onto the surface of the magnetic nanoparticles via electrostatic interactions, with the nanoparticles serving as the core and the outermost shell covered with gemcitabine (GEM), a drug delivery carrier used in the treatment of pancreatic cancer (Figure 1).

## 2. Results and Discussion

### 2.1. Characterization of the Particle Size and Potential of LBL MNPs

The physical and chemical properties of nanocarriers are crucial for their cellular internalization. For example, magnetic nanoparticle (MNP) sizes between 50 and 200 nm are more likely to accumulate at tumor sites through the enhanced permeability and retention (EPR) effect and be internalized by tumor cells [20]. Dynamic light scattering (DLS) was employed to determine the particle size, polydispersity index (PDI), and zeta potential (surface charge) of the LBL MNPs with varying numbers of layers.

As shown in Figure 2, the particle sizes of LBL MNPs with different numbers of layers were measured to reflect electrostatic interactions between the positively charged polcation and negatively charged polyanion. This change in particle size is likely due to the introduction of the negatively charged polymer, poly4-styrenesulfonate (PSS), and the positively charged polyallylamine (PAH). Simultaneously, the particle size of the MNPs increased as each polymer layer was added, confirming the successful preparation of LBL MNPs (Figure 2a). Specifically, the particle sizes of the polymers in the 5-, 7-, and 9-layer configurations were 157.6 ± 65.7 nm, 198.7 ± 24.2 nm, and 369.0 ± 199.1 nm, respectively, and the PDIs were 1.2 ± 1.6, 0.8 ± 0.5, and 1.22 ± 1.2, respectively. Additionally, the zeta potentials for the 5-, 7-, and 9-layer polymers were −38.0 ± 17.4 mV, −15.6 ± 5.5 mV, and −50.7 ± 13.9 mV, respectively, all of which were negative, facilitating subsequent electrostatic loading of the positively charged GEM (Figure 2b). Transmission electron microscopy (TEM) images of the 1- and 7-layer coated LBL MNPs were also acquired to visualize the polymer layers (Figure 2c,d).

### 2.2. Heat Generation Behavior of LBL MNPs

Magnetic hyperthermia, which utilizes MNPs as a heat source, is a technique that transfers heat to tumor cells. This method generates heat through the interaction with alternating magnetic fields (AMF), making it an effective treatment for targeting deep-seated, inaccessible tumors. Heat generation occurs when magnetic nanoparticles are exposed to a magnetic field, with energy being converted to heat during relaxation processes based on the Néel and Brownian relaxation mechanisms. Therefore, it is essential to use AMF rather than direct-current magnetic fields to ensure sustained heat generation. The heat-generation behavior of polymer-coated LBL MNPs under AMF exposure was observed, as shown in Figure 3. When the MNPs were subjected to AMF for 240 s, the temperature of the LBL MNPs suspension increased from 29.0 to 42.3 °C. This temperature increase is similar to that of bare MNPs. Hyperthermia requires a temperature range of 42–45 °C, and the heat generated by LBL MNPs falls within this range, making them suitable for hyperthermia applications. Eddy currents induced by magnetic fields can affect tissues and organs, potentially causing carbonization and necrosis. However, the frequency used in this study was only 281 kHz, which is well below the safe frequency threshold for AMF (100–300 kHz) [21].

### 2.3. Relationship Between Cell Killing Effect of LBL MNPs, Concentration, and Layer Number

The drug release curves of GEM encapsulated in LBL MNPs are shown in Figure 4a. GEM loading increased with the number of layers, with a release of 7.61 µg/mL GEM after 24 h in a 100 µg/mL suspension of 7-layer particles (Figure 4a). When PANC-1 cells were treated with LBL MNPs of different layer numbers, a concentration-dependent increase in cytotoxicity was observed for each layer (Figure 4b). Regarding the drug release behavior, GEM on the LBL particles was electrostatically bound; therefore, it is likely that it was gradually released by displacement with other electrolytes. Results from the two-way analysis of variance (ANOVA) followed by a Bonferroni post-hoc test indicate that both the number of layers (F (3, 48) = 13.349, *p* < 0.001, η^2^ = 0.455) and concentration (F (5, 48) = 151.989, *p* < 0.001, η^2^ = 0.941) significantly affected survival. Additionally, a significant interaction effect between layer number and concentration was observed (F (15, 48) = 2.881, *p* = 0.003, η^2^ = 0.474), suggesting that the concentration effect on survival varies depending on the layer type. Post-hoc Bonferroni tests revealed significant differences in survival across different concentration levels (*p* < 0.001). Higher concentrations resulted in significantly lower survival rates, with a steep decline from 0 to 100 µg/mL. Among the different layers, 7 layers showed the most distinct behavior, exhibiting significantly lower survival rates compared to the 5, 9, and 11 layers (*p* < 0.001). The differences between the 9 and 11 layers were not statistically significant (Figure 4b). The superior performance of 7-layer LBL MNPs may be attributed to an optimal balance between drug-loading capacity and controlled release. In addition, the significant interaction effect (*p* = 0.003) suggests that the impact of concentration on survival varies depending on the layer structure. Notably, 7 layers exhibited the most pronounced decrease in survival, indicating that the specific structural or compositional properties of this layer may enhance drug penetration or toxicity. Studies have shown that the number of layers in multilayered nanoparticles directly affects drug release profiles and cellular uptake efficiency [22,23]. A 7-layer configuration likely provides sufficient drug-loading sites while maintaining structural integrity, which facilitates efficient cellular internalization. In contrast, both under-structured (e.g., the 5-layer) and over-structured (e.g., 9- and 11-layer) MNPs may exhibit suboptimal performance. In particular, the zeta potential of the 5- and 9-layer MNPs showed a relatively strong negative surface charge, suggesting difficulty in GEM release. On the other hand, over-structured polymers may hinder drug release due to excessive layering. These studies demonstrate that the number of nanoparticle layers significantly affects drug release profiles and cellular uptake efficiency, supporting the idea that 7-layer LBL nanoparticles strike an optimal balance.

### 2.4. Enhancement of the Synergistic Cell-Killing Effect by LBL MNPs and AMF Application

Combining 7-layer LBL MNPs with magnetic hyperthermia confirmed that the inhibition of cell proliferation activity was dependent on both the LBL MNPs concentration and AMF application time. Combining 100 µg/mL LBL MNPs with 30 min of AMF treatment resulted in approximately 98% inhibition of cell proliferation compared to the absence of LBL MNPs. Similarly, the combined use of LBL MNPs and AMF significantly increased the number of test cells in a concentration- and time-dependent manner. A two-way ANOVA revealed significant main effects of LBL MNP concentration (η^2^ = 0.997 for viability, η^2^ = 0.952 for cell death) and AMF application time (η^2^ = 0.975 for viability, η^2^ = 0.913 for cell death) on cell viability and cell death induction. A significant interaction between these factors was observed for both viability (η^2^ = 0.923, *p* < 0.001) and cell death (η^2^ = 0.811, *p* < 0.001), indicating that AMF enhances the cytotoxic effects of LBL MNPs. Post hoc analysis revealed a dose-dependent decrease in viability, with significant reductions observed starting from 10.00 μg/mL LBL MNPs and 5.00 min of AMF application. However, a plateau effect was noted at 30.00 µg/mL and above for LBL MNPs and at longer AMF application times, suggesting a saturation of cytotoxic effects. Similarly, cell death significantly increased with LBL MNP concentration and AMF application, showing a synergistic effect between these two factors (Figure 5). These findings suggest that AMF exposure enhances LBL MNPs-induced cytotoxicity, likely through increased intracellular nanoparticle uptake and hyperthermia effects [24]. The observed saturation effect is consistent with previous studies showing that beyond a threshold, additional drug exposure does not significantly enhance apoptosis [25]. Similarly, the plateau effect at higher AMF intensities aligns with reports that hyperthermia-induced cytotoxicity has an upper limit [26]. The synergistic interaction supports previous findings that magnetic hyperthermia enhances chemotherapy by increasing reactive oxygen species (ROS) generation and mitochondrial damage [27]. LBL MNPs induce a local temperature rise through AMF application, inducing apoptosis and necrosis of tumor cells [28,29]. In this study, cell proliferation inhibition and cell death induction were enhanced with increasing LBL MNPs concentration and AMF irradiation time, which is thought to be related to the heating effect of LBL MNPs by AMF. In particular, the inhibition of cell proliferation reached approximately 98% after combining 100 µg/mL LBL MNPs with 30 min of AMF application, indicating that the combination of an appropriate LBL MNPs concentration and AMF application time may maximize the therapeutic effect. Hyperthermia induced by MNPs occurs because temperature increases the permeability of tumor cells and their sensitivity to chemotherapy [30]. These results suggest that treatment with LBL MNPs exposed to AMF made cancer cells more sensitive to chemotherapy and thus efficiently induced cell death [30].

### 2.5. Enhancement of the Synergistic Cell-Killing Effect by LBL MNPs, AMF, and X-Ray Irradiation

A combination of radiation therapy and chemotherapy can enhance the therapeutic effect and is widely used in clinical practice. The synergistic effect of these therapies is expected to result in better proliferation inhibition of cancer cells and more efficient treatment. GEM, in addition to its cytotoxic effects, is also a potent radiosensitizer. Such nucleotide derivatives are attractive when used in combination with radiotherapy. Due to their preferential cytotoxicity against proliferating cells, these derivatives may also reduce cancer cell proliferation and slow cellular repopulation during radiotherapy (RT). However, hypoxic cancer cells are radio-tolerant but sensitive to heat [31]. We confirmed the enhancement of the cell-killing effect by adding X-ray irradiation to the LBL MNPs treatment and AMF application. As a result, significant inhibition of cell proliferation was confirmed compared to the combined use of the two treatments. In particular, in the group where AMF was applied for 30 min, an enhancement effect was confirmed that inhibited proliferation activity by more than 80% even in the low concentration (5 µg/mL) LBL MNPs group, and an enhancement effect of more than 95% was confirmed in the high concentration (100 µg/mL) group. Similarly, the induction of cell death was confirmed using a combination of these three treatments. The combination of LBL MNPs and 5 Gy X-rays also showed a maximum cell death induction effect of approximately 3.7 times compared to the control group. The effect was enhanced depending on the LBL MNPs concentration and AMF application time, and the combination of 100 µg/mL LBL MNPs, 30 min of AMF application, and X-rays induced cell death 13 times higher than that of the control (Figure 5b). A three-way ANOVA revealed significant main effects of X-ray irradiation (η^2^ = 0.967 for viability, η^2^ = 0.933 for cell death), LBL MNPs concentration (η^2^ = 0.996 for viability, η^2^ = 0.933 for cell death), and AMF intensity (η^2^ = 0.805 for viability, η^2^ = 0.791 for cell death). Importantly, significant interaction effects were observed for X-ray and drug concentration (η^2^ = 0.924 for viability, η^2^ = 0.875 for cell death), X-ray and AMF (η^2^ = 0.805 for viability, η^2^ = 0.550 for cell death), and drug concentration and AMF (η^2^ = 0.923 for viability, η^2^ = 0.712 for cell death). A three-way interaction (η^2^ = 0.881 for viability, η^2^ = 0.551 for cell death, *p* < 0.001) was also observed, indicating that combining all three factors results in the greatest cytotoxic effect. Post hoc analyses revealed that X-ray irradiation significantly enhances the effects of both LBL MNPs and AMF (Figure 6). Although only LBL MNPs and AMF reduce viability and increase cell death, the addition of X-rays further amplifies these effects, likely due to increased DNA damage and apoptosis induction. The strong interaction effects suggest that these treatments work synergistically rather than additively. These results are consistent with prior findings that radiotherapy induces DNA double-strand breaks, enhancing chemotherapy-induced apoptosis [32]. Furthermore, AMF exposure has been shown to enhance nanoparticle-mediated hyperthermia, which can increase ROS generation and further sensitize cancer cells to radiation [33]. The significant three-way interaction suggests that LBL MNPs, AMF, and X-ray irradiation together create a potent combination for maximizing cancer cell death [34]. It has been reported that hyperthermia using LBL MNPs and AMF enhances the therapeutic effect on tumor cells when combined with radiotherapy [35]. DNA double-strand breaks are a major cause of radiation-induced cell death. Nucleotide derivatives such as GEM, as DNA synthesis inhibitors, have the potential to inhibit the repair of radiation-induced DNA damage. Once incorporated into the DNA repair patch, nucleoside analogs may trigger an apoptotic response similar to that observed during replication. DNA damage is induced by radiation at all stages of the cell cycle, where the cell cycle enters the S phase arrest mode, and cells in the S phase are most sensitive to heat [36]. This mechanism suggests the possibility of extending the cytotoxicity of these analogs to cells outside the S phase [31]. In this study, cell proliferation inhibition and cell death induction were significantly enhanced by combining LBL MNPs and AMF with X-ray irradiation. In particular, even at a low concentration (5 µg/mL) of LBL MNPs, the inhibition of cell proliferation reached 80% or more after 30 min of AMF application and X-ray irradiation, indicating the potential for achieving a high therapeutic effect while reducing the concentration of LBL MNPs. In addition, the cell death induction effect reached 13 times that of the control group by combining a high concentration (100 µg/mL) of LBL MNPs, 30 min of AMF application, and 5 Gy of X-ray irradiation, suggesting a strong synergistic effect from the combination of all three treatments (Figure 5 and Figure 6).

Surgery, chemotherapy, and radiation therapy are the most common cancer treatments currently available, with several new therapies under development [37,38,39]. Drug delivery systems (DDS) enhance the therapeutic efficiency of chemotherapy. The MNPs presented here can be used in combination with chemoradiotherapy and hyperthermia, in addition to delivering high concentrations of anticancer drugs locally. Furthermore, if several recently developed chemotherapeutic agents [40,41] can also be encapsulated in these MNPs, it would be possible to create a therapy that enhances antitumor effects while reducing systemic side effects. Research is actively being conducted with this goal in mind.

This study had several limitations and challenges. The MNPs tended to aggregate during layering, as shown by changes in particle size and electron microscopic images. Maintaining an appropriate particle size is crucial for the therapeutic effect of DDS [42]. A particle design that prevents aggregation as the number of layers increases is needed to ensure the particle size matches the tumor characteristics. Additionally, to load more anticancer agents, a technique for coating multiple layers of anticancer drugs must be developed.

These results suggest that the combination of hyperthermia therapy and radiation therapy using LBL MNPs could be a novel treatment strategy for intractable tumors such as pancreatic cancer. Optimizing the concentration of LBL MNPs and the duration of AMF exposure may maximize the therapeutic effect and minimize side effects. In the future, in vivo efficacy verification and safety evaluations are expected to be promoted, paving the way for further clinical application. The results of this study indicate that the combination of LBL MNPs, AMF, and radiation therapy provides a powerful therapeutic effect on tumor cells and is gaining attention as a new option for future cancer treatment.

## 3. Materials and Methods

### 3.1. Preparation of LBL Nanoparticles and LBL Nanoparticles Loaded with GEM

The LBL nanoparticles were prepared using electrostatic adsorption. Briefly, positively charged MNPs (EMG 607, Ferrotec Material Technologies Corporation, Tokyo, Japan) and negatively charged PSS (5 mg/mL, 516-85751, FUJIFILM Wako Pure Chemical Corporation, Osaka, Japan) were mixed at a ratio of 1:49 for 15 min, placed in a neodymium magnet for 10 min, and centrifuged at 4 °C (15,000 rpm, 15,850 rcf) for 15 min to remove the supernatant. The precipitate was washed with Milli-Q water and centrifuged twice to remove unbound PSS, obtaining the first layer of LBL nanoparticles. Subsequently, the precipitate was resuspended in a small amount of double-distilled water, shaken, and mixed with PAH (5 mg/mL, 283223, Merck KGaA, Darmstadt, Germany) at a ratio of 1:40 for 15 min. As described above, after washing with Milli-Q water and centrifuging twice, the solution was repeatedly mixed with the PSS solution to prepare LBL MNPs with a different number of layers.

For LBL MNPs loaded with GEM (G-0367-1G, Tokyo Chemical Industry Co., Ltd., Tokyo, Japan) after preparing the seventh layer of LBL MNPs, negatively charged LBL MNPs and positively charged GEM were electrostatically adsorbed to prepare LBL MNPs containing GEM. Specifically, GEM (5 mg/mL) and LBL MNPs were mixed, shaken for 15 min, and then centrifuged at 15,000 rpm (15,850 rcf) for 15 min at 4 °C to remove unbound GEM. The mixture was washed with Milli-Q water and centrifuged twice (15,000 rpm, 15,850 rcf, 15 min, 4 °C) to obtain LBL MNPs containing GEM.

### 3.2. Characterization of LBL Nanoparticle and Size Distribution and Zeta Potential Measurements

The particle size and zeta potential of the LBL nanoparticles were measured using dynamic light scattering (DLS; Brookhaven Instrument Corp., Holtsville, NY, USA) and a Zetasizer (Malvern Instrument Ltd., Worcestershire, UK), respectively. All particle size measurements were performed using a He-Ne laser beam at 658 nm and a scattering angle of 90°. The final micelle concentration was 1 mg/mL. For each sample, the data obtained from three measurements were averaged to yield the mean particle size and zeta potential.

### 3.3. Heat Generation Behavior of LBL MNPs upon AMF

The heating profiles of LBL MNPs with and without were investigated by placing them in the center of a copper coil and applying an AMF (HOSHOT2, Alonics Co., Ltd., Tokyo, Japan). Heat was generated by the AMF (1.1 kA/m, 281 kHz, 4.0 V). The heating profiles were recorded by taking photos every 20 s using a FLIR thermal camera (CPA-E6, FLIR systems Japan K.K., Tokyo, Japan).

### 3.4. Cell Culture

The human pancreatic cell line PANC-1 (RCB2095; RIKEN Bioresearch Center, Tsukuba, Japan) was cultured in Ham’s F-12 medium (087-08335, FUJIFILM Wako Pure Chemical Corporation, Osaka, Japan) containing 10% fetal bovine serum (FBS; Lot. 2404079, Thermo Fisher Scientific K.K., Tokyo, Japan) and antibiotics (100 U/mL penicillin and 100 µg/mL streptomycin; Merck KGaA, Darmstadt, Germany) in a 5% CO_2_ incubator at 37 °C. Cells were passaged twice per week, and treatment samples were prepared three days before AMF treatment.

### 3.5. Cytotoxic Effects of LBL MNPs Alone

The cytotoxicity of LBL MNPs alone was evaluated using 5-, 7-, 9-, and 11-layer LBL MNPs. PANC-1 cells were seeded at 1 × 10^4^ cells/500 µL/well in a 24-well plate (3526, Corning, NY, USA) two days before LBL MNP treatment. The following day, the cells were treated with LBL MNPs at concentrations of 0, 1, 3, 10, 30, and 100 µg/mL for 24 h. After 24 h, cell viability was assessed using the CellTiter-Glo 2.0 assay (G9241, Promega, Madison, WI, USA) according to the manufacturer’s protocol. The assays were conducted in triplicate, and luminescence signals were measured using a microplate reader (Infinite 200 PRO, TECAN, Männedorf, Switzerland). To minimize potential interference from LBL MNPs in fluorescence-based assays, the medium was replaced with a fresh culture medium before measurement. Specifically, after 24 h of incubation with LBL MNPs, the culture medium containing the nanoparticles was removed, and the cells were washed with phosphate-buffered saline (PBS) (166-23555, FUJIFILM Wako Pure Chemical Corporation, Osaka, Japan). Fresh culture medium was then added before performing cell viability and cytotoxicity assays. This procedure ensured that the fluorescence signals were not affected by nanoparticle-induced scattering or absorption effects.

### 3.6. Cytotoxic Effects Using LBL Nanoparticles Combined with AMF and/or X-Rays

Using the 7-layer LBL MNPs, which were the most effective in the previous experiment, we evaluated the cytotoxicity of LBL MNPs combined with AMF and X-rays. Two days before LBL MNPs treatment, PANC-1 cells were seeded at 2 × 10^4^ cells/300 µL/well in a µ-Dish 35 mm Quad (80416, ibidi GmbH, Gräfelfing, Germany) with four seeding areas. The cells were treated with 7-layer LBL MNPs at concentrations of 0, 2, 10, and 50 µg/mL for 24 h. The cells were then subjected to AMF (1.1 kA/m, 281, and 4.0 V) for 0, 10, and 30 min. The cells were subsequently irradiated with 5 Gy X-rays at 130 kV, 5 mA, a dose rate of 0.5 Gy/min, and room temperature using an X-ray generator RX-650 (Faxitron Bioptics LLC, Tucson, AZ, USA) installed at the Proton Medical Research Center, University of Tsukuba. Twenty-four hours after irradiation, cell proliferation activity was evaluated using Celltiter-Glo 2.0 (G9241, Promega, Madison, WI, USA), and changes in the number of dead cells were evaluated using the CellTox Green Cytotoxicity Assay (G8742, Promega, Madison, WI, USA) according to the manufacturer’s protocol. The assays were conducted in triplicate, and luminescence signals were measured using a microplate reader (Infinite 200 PRO, TECAN, Männedorf, Switzerland). Regarding cell proliferation activity and the increase in the number of dead cells, the relative light unit (RLU) values obtained from the plate reader were normalized with the LBL MNPs concentration of 0 µg/mL and AMF application time of 0 min as the standard.

### 3.7. Statistical Analysis

All statistical analyses were performed using IBM SPSS Statistics (version 29.0.2.0, IBM Corp., Armonk, NY, USA). Data are expressed as the mean ± standard error of the mean (SE) obtained from three independent experiments. A two- or three-way ANOVA was conducted to assess the effects of layer, concentration, and X-ray exposure on cell viability. Interaction effects were also analyzed. If significant effects were detected, multiple comparisons were performed using the Bonferroni correction to control for Type I error. A *p*-value of < 0.05 was considered statistically significant.

## 4. Conclusions

LBL MNPs have been developed for combination chemotherapy, thermotherapy, and RT. To deliver the anticancer drug GEM, the surface charge of the MNPs was controlled through the electrostatic interaction between PSS and PAHs, which was designed to bind GEM to the outermost shell. When the cytotoxicity of GEM-loaded LBL MNPs was examined, particles with 7 layers of PSS/PAHs showed the most effective cytotoxicity. This is because the surface charge is precisely controlled by the LBL, and GEM is released gradually through exchange with the body’s electrolytes. In addition, AMF irradiation was found to induce fever, and the combination of chemotherapy, hyperthermia, and RT showed synergistic cytotoxicity. This may be because hyperthermia and RT complement each other’s limitations, and GEM is most effective when the cell cycle is arrested in the S phase, where DNA damage is difficult to repair [24,26]. Beyond conventional chemotherapy and RT, the delivery of drugs and heat sources that synergize these treatments is expected to drive the development of more efficient combination therapies.

## Figures and Tables

**Figure 1 molecules-30-01382-f001:**
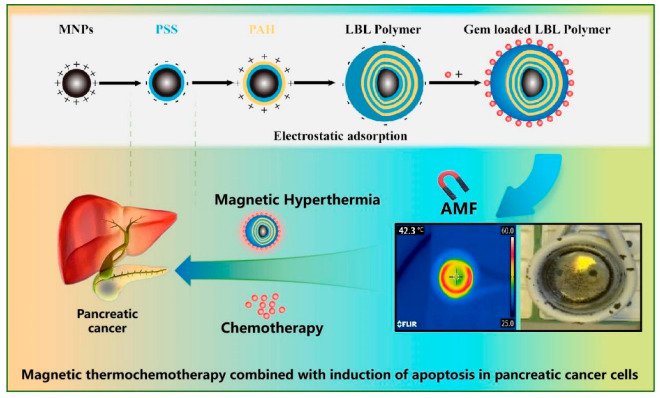
Schematic of layer-by-layer nanoparticle preparation combined with thermal chemoradiotherapy for efficient synergistic treatment of pancreatic cancer.

**Figure 2 molecules-30-01382-f002:**
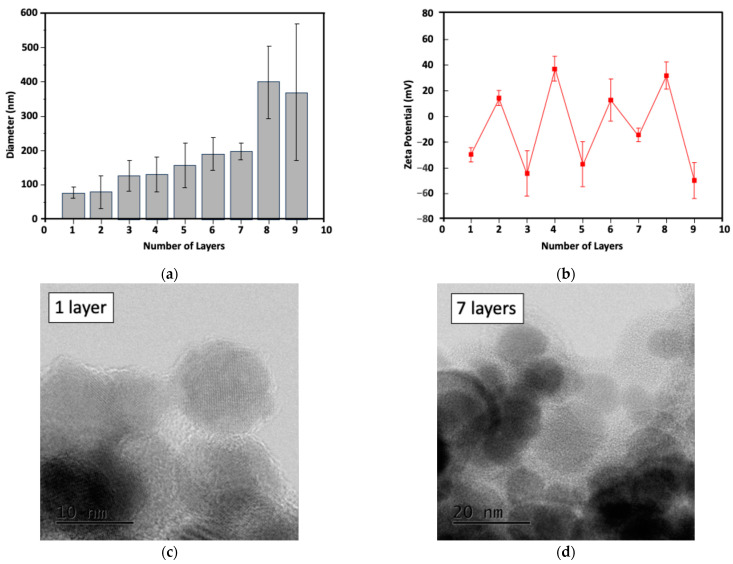
Changes in particle size (**a**) and zeta potential (**b**) for LBL MNPs. TEM images of LBL MNPs with 1- (**c**) and 7- (**d**) layers.

**Figure 3 molecules-30-01382-f003:**
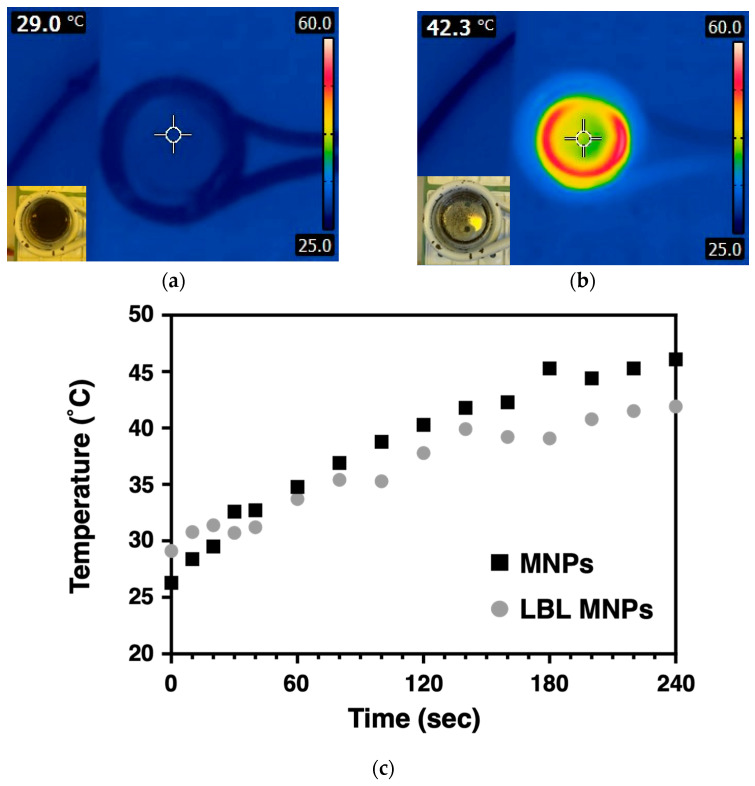
Heat generation behavior of LBL MNPs coated with polymers when exposed to an AMF field. Heatmap images were obtained when particles were exposed for (**a**) 0 and (**b**) 4 min. MNPs coated with up to 7 layers of polymer MNP suspension in water are shown. (**c**) Data showing the temperature increase per min for LBL MNPs (●) and MNPs (■).

**Figure 4 molecules-30-01382-f004:**
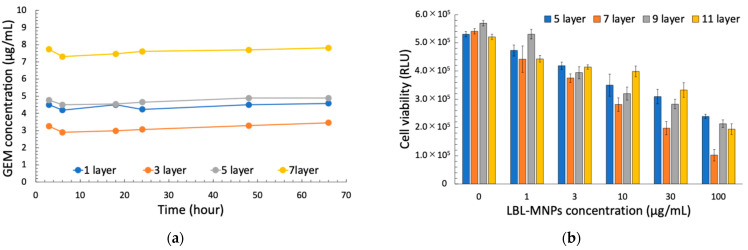
(**a**) Drug release profile of layer-by-layer magnetic nanoparticles (LBL MNPs) and (**b**) the effect of LBL MNPs on cell viability. The concentration of released GEM (µg/mL) was measured over time (h) for nanoparticles with different numbers of layers (1, 3, 5, and 7). GEM release showed a layer-dependent profile, with nanoparticles having more layers exhibiting a more sustained release. The data indicate that increasing the number of layers results in prolonged drug release. Cell viability (relative luminescence units, RLU) was assessed after treatment with LBL MNPs at various concentrations (0, 1, 3, 10, 30, and 100 µg/mL). Different numbers of layers (5-, 7-, 9-, and 11-layer) were evaluated for their cytotoxic effects. Data are presented as the mean ± standard error (SE) from three independent experiments (n = 3). Cell viability decreased with increasing LBL MNP concentration, with variations observed depending on the number of layers.

**Figure 5 molecules-30-01382-f005:**
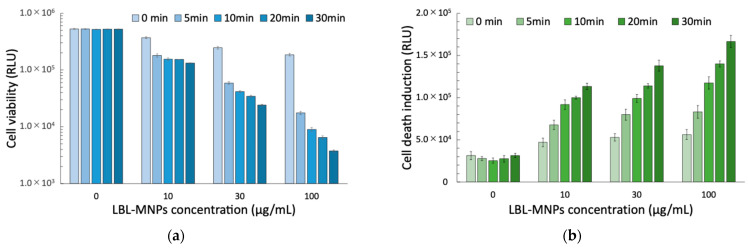
Effects of layer-by-layer magnetic nanoparticles (LBL MNPs) on (**a**) cell viability and (**b**) cell death induction. (**a**) Cell viability and (**b**) cell death induction were assessed after treatment with 7-layer LBL-MNPs at various concentrations (0, 10, 30, and 100 µg/mL) and different alternating magnetic field (AMF) exposure times (0, 5, 10, 20, and 30 min). After 24 h of treatment, cell viability decreased in a concentration- and time-dependent manner, whereas cell death induction increased accordingly. Data are presented as mean ± standard error (SE) from three independent experiments (n = 3).

**Figure 6 molecules-30-01382-f006:**
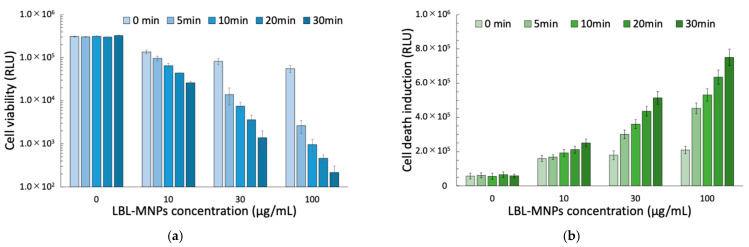
Effects of layer-by-layer magnetic nanoparticles (LBL MNPs) on (**a**) cell viability and (**b**) cell death induction. (**a**) Cell viability and (**b**) cell death induction were assessed after treatment with 7-layer LBL MNPs at various concentrations (0, 10, 30, and 100 µg/mL) and different alternating magnetic field (AMF) exposure times (0, 5, 10, 20, and 30 min) and X-ray irradiation (5 Gy). After 24 h of treatment, (**a**) cell viability decreased in a concentration- and time-dependent manner, whereas (**b**) cell death induction increased accordingly. Data are presented as mean ± standard error (SE) from three independent experiments (n = 3).

## Data Availability

The data supporting the findings of this article are available from the corresponding author, Y.M., upon reasonable request.

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
