# Peer review of "Development of Layer-by-Layer Magnetic Nanoparticles for Application to Radiotherapy of Pancreatic Cancer"

_molecules, 2025, doi:10.3390/molecules30061382_

Round 1

Reviewer 1 Report

Comments and Suggestions for Authors

1. The Figure 2 should be separated by the size and Zeta.

2. Please also give the SEM and TH-TEM for the LBL MNPs

3. why not put it in the control group?

4. How about it is the Synergistic effect? I can not identify from the current results.

5.  Some work on this topic should be addressed, including Small, 2021, 17, 2007486; Small, 2020, 16, 2004173. And Adv. Healthc. Mater., 2023, 12, 2202911

6. What is the sample number of each group in the cell experiment? The data of cytocompatibility experiments need to be provided.

Comments on the Quality of English Language

revision carefully, typesetting work could be checked.

Author Response

Thank you for your instructions. We have prepared a point-by-point response to the reviewers’ comments and have uploaded it as an attachment. Please see the attachment file “Respons_to_reviewer1" for details.

Reviewer 2 Report

Comments and Suggestions for Authors

The article authored by Nobuyoshi Fukumitsu describes the use of loaded layer-by-layer magnetic nanoparticles (LBL MNPs) for the treatment of pancreatic cancer. LBL MNPs were loaded with gemcitabine and tested on PANC-1 cells. The results clearly demonstrated that combination of chemotherapy, hyperthermia therapy and radiation therapy could be useful in the treatment of pancreatic cancer. The article is clearly written and the results are described thoroughly. In my opinion the article deserves to be published without modifications.

Author Response

Thank you for your instructions. We have prepared a point-by-point response to the reviewers’ comments and have uploaded it as an attachment. Please see the attachment file “Respons_to_reviewer2" for details.

Reviewer 3 Report

Comments and Suggestions for Authors

Interesting and topical paper; however, key conclusions (e.g., significant decrease in the number of cells) are not supported by the results. Some controls are missing. The number of issues is significant; therefore, I suggest that this paper is not suitable for publication in its present form.

1. The title of the manuscript does not reflect its content. The treatment of pancreatic cancer was not performed in this study. Instead, the paper focuses on the cytotoxicity of nanoparticles toward a tumor cell line under specific experimental conditions.

2. The statistical analysis used in the manuscript is inadequate. The only statistical method employed is a T-test without any corrections (as stated in section 3.7). This approach is unsuitable for comparing multiple groups, as presented in Figures 4, 5, and 6. A correction for multiple comparisons (e.g., Bonferroni correction), ANOVA, or another appropriate statistical method should be applied. In its current form, the statistical assessment of the data is likely incorrect.

3. The source and structure of the magnetic nanoparticles are not described. This information is critical for reproducibility and understanding the material properties.

4. Line 229 - The preparation method of "positively charged LBL MNP" is not described. Additionally, the concentration of nanoparticles used is not specified. Without these details, the method cannot be independently reproduced.

5. Gemcitabine loading was not confirmed experimentally. It is unclear whether gemcitabine was successfully loaded onto the nanoparticles. Was this verified?

6. Line 116 - The statement regarding drug release behavior ("With respect to the drug release behavior, GEM on the LBL particles was electrostatic") is unsubstantiated. A drug release profile is not shown in the manuscript.

7. Potential interference in cell viability assays was not addressed. The authors used a luminescent assay to measure cell viability. However, magnetic nanoparticles at concentrations of 10–100 µg/mL typically form relatively dark dispersions and scatter light. Did the authors account for or test nanoparticle interference in this assay?

8. Figures 5 and 6 lack sufficient explanation in their legends. The legends should provide clear and detailed descriptions to allow readers to interpret the data accurately.

9. Figures 5b and 6b present "relative number" of dead cells, which is an unusual parameter to report. Typically, the percentage of live cells is more informative, as the goal of cancer therapy is to eliminate all cancer cells. Reporting an increase in dead cells alone may be misleading since a small percentage of dead cells (e.g., 1–5%) is present in most cell cultures under baseline conditions. A 5–6-fold increase in this small fraction does not necessarily indicate a significant therapeutic effect if most cells (70–94%) remain viable. The percentage of live cells should be specified instead.

10. Control experiments are missing in Figure 6. Specifically, controls such as X-ray exposure alone and MNP without GEM combined with X-ray are absent. These controls are essential to determine whether gemcitabine is necessary for the observed effects.

11. Line 251 - "Micelles" are mentioned but not explained or defined in the context of this study. This term requires clarification.

12. Lines 254–258 - There are formatting issues in this section that need correction.

13. Abbreviations (e.g., GEM/LBL) are inconsistently used throughout the manuscript. For example, this abbreviation appears only in section 3.1 and is not used elsewhere in the text. It remains unclear whether GEM-loaded or unloaded nanoparticles are being referred to in other sections.

14. Centrifugation rate should be expressed in relative centrifugal force (rcf). Providing centrifugation speed solely in rpm is insufficient, as rcf accounts for rotor dimensions and provides a standardized measure.

Author Response

Thank you for your instructions. We have prepared a point-by-point response to the reviewers’ comments and have uploaded it as an attachment. Please see the attachment file “Respons_to_reviewer3" for details.

Round 2

Reviewer 1 Report

Comments and Suggestions for Authors

accept